# Concordance between Antimicrobial Resistance Phenotype and Genotype of *Staphylococcus pseudintermedius* from Healthy Dogs

**DOI:** 10.3390/antibiotics11111625

**Published:** 2022-11-15

**Authors:** Joaquim Viñes, Norma Fàbregas, Daniel Pérez, Anna Cuscó, Rocío Fonticoba, Olga Francino, Lluís Ferrer, Lourdes Migura-Garcia

**Affiliations:** 1Vetgenomics, Edifici EUREKA, PRUAB, Campus de la Universitat Autònoma de Barcelona (UAB), Bellaterra, 08193 Barcelona, Spain; 2Molecular Genetics Veterinary Service (SVGM), Universitat Autònoma de Barcelona (UAB), Bellaterra, 08193 Barcelona, Spain; 3Department of Animal Medicine and Surgery, Universitat Autònoma de Barcelona (UAB), Bellaterra, 08193 Barcelona, Spain; 4Institute of Science and Technology for Brain-Inspired Intelligence, Fudan University, Shanghai 200433, China; 5Unitat mixta d’Investigació IRTA-UAB en Sanitat Animal, Centre de Recerca en Sanitat Animal (CReSA), Campus de la Universitat Autònoma de Barcelona (UAB), Bellaterra, 08193 Barcelona, Spain; 6Institut de Recerca i Tecnologia Agroalimentàries (IRTA), Programa de Sanitat Animal, Centre de Recerca en Sanitat Animal (CReSA), Campus de la Universitat Autònoma de Barcelona (UAB), Bellaterra, 08193 Barcelona, Spain

**Keywords:** *Staphylococcus pseudintermedius*, antibiotic resistance, nanopore sequencing, phenotype-genotype concordance, healthy dog

## Abstract

*Staphylococcus pseudintermedius*, a common commensal canine bacterium, is the main cause of skin infections in dogs and is a potential zoonotic pathogen. The emergence of methicillin-resistant *S. pseudintermedius* (MRSP) has compromised the treatment of infections caused by these bacteria. In this study, we compared the phenotypic results obtained by minimum inhibitory concentration (MICs) for 67 *S. pseudintermedius* isolates from the skin of nine healthy dogs versus the genotypic data obtained with Nanopore sequencing. A total of 17 antibiotic resistance genes (ARGs) were detected among the isolates. A good correlation between phenotype and genotype was observed for some antimicrobial classes, such as ciprofloxacin (fluoroquinolone), macrolides, or tetracycline. However, for oxacillin (beta-lactam) or aminoglycosides the correlation was low. Two antibiotic resistance genes were located on plasmids integrated in the chromosome, and a third one was in a circular plasmid. To our knowledge, this is the first study assessing the correlation between phenotype and genotype regarding antimicrobial resistance of *S. pseudintermedius* from healthy dogs using Nanopore sequencing technology.

## 1. Introduction

*Staphylococcus pseudintermedius* is a common bacteria found in the mucous membranes and skin of dogs. Carriage of *S. pseudintermedius* in healthy dog populations can range between 46% and 92% [1,2,3]. However, it is also the primary pathogen isolated from clinical canine specimens, especially from pyoderma, otitis externa, and superficial folliculitis. In recent decades, the emergence of methicillin-resistant *S. pseudintermedius* (MRSP) has compromised the treatment of these infections in veterinary medicine. Furthermore, in recent years, several studies have reported community-acquired infections with *S. pseudintermedius* transmitted from dogs to humans directly or by contamination of the household environment [4,5], highlighting their zoonotic potential. As a result, *S*. *pseudintermedius* has been identified among the most relevant antimicrobial-resistant (AMR) bacteria in the European Union for dogs and cats, and its eligibility to enter the list of animal diseases within the framework of the Animal Health Law has been recently discussed [6].

To date, the vast majority of studies have focused on *S. pseudintermedius* isolated from diseased dogs and have evidenced a high level of resistance to clinically important antimicrobials, such as beta-lactams, lincosamides, and fluoroquinolones [7]. In many cases, the presence of the *mecA* gene, or its homolog, *mecC* producing a modified penicillin-binding protein (PBP), provides them with phenotypic resistance to all beta-lactam antimicrobials registered in veterinary medicine, limiting the therapeutic options. Additionally, the global distribution of some sequence types (ST) associated with MRSP causing infections, such as ST71 in Europe and ST68 in North America, have contributed to unravel the epidemiology of this pathogen and identifying lineages of *S. pseudintermedius* with pathogenic and zoonotic potential [8].

However, to have a broader knowledge of the epidemiology of this opportunistic pathogen and to help veterinarians guide antimicrobial stewardship, it is necessary to study not only isolates associated with clinical cases but also the commensal population inhabiting the skin of healthy dogs. Furthermore, strain diversity within each individual animal has been studied by traditional methods, such as phenotypic tests, pulsed-field gel electrophoresis, or PCR, describing high genotypic diversity between different anatomical sites of the animal [9] but not many differences in antimicrobial resistant profiles. Hence, high-throughput sequencing technologies have the advantage of studying complete genomes. In particular, long-read sequencing technologies, such as Oxford Nanopore Technology (ONT), offer the advantages of closing genomes and plasmids, enabling establishment of the location and the genetic context of the different antimicrobial resistance genes [10,11]. This facilitates the assessment of the risk of transmission being higher when it is plasmid-borne and has the potential of being horizontally transferred intra and inter-species.

To our knowledge, whole genome sequencing techniques have not been fully applied to determine the prevalence and antimicrobial resistance determinants within *S. pseudintermedius* of healthy dogs. Therefore, the aim of this study was to characterize genotypically and phenotypically the population of *S. pseudintermedius* from the skin of different anatomical areas of healthy dogs and to correlate the phenotype with the genotype. Additionally, the location of the different antimicrobial resistant genes and their genetic context have been identified.

## 2. Results

### 2.1. Distribution of MRSP and MDR Strains

Sixty-seven *S. pseudintermedius* were recovered from the skin of nine healthy dogs. The isolates were obtained from four different anatomical areas (Table 1): 27 from the perioral (six dogs), 16 from the perianal (six dogs), 13 from the inguinal (four dogs), and 11 from the nasal region (five dogs).

As seen in Figure 1a, minimal inhibitory concentration assays (MIC) described resistance to penicillin in 82.1% of the isolates, to tetracycline in 71.6%, to ampicillin in 59.7%, to oxacillin in 50.8%, to erythromycin and clindamycin in 43.3%, to ceftriaxone and ciprofloxacin in 37.3%, to co-trimoxazole in 32.8%, to gentamicin in 10.5%, to vancomycin in 1.5%, and no isolates were resistant to rifampicin. Multidrug resistance was described in 43.3% of the isolates. The most common combination of resistances among multidrug resistant isolates (MDR) was the resistance to erythromycin, penicillin, ampicillin, ceftriaxone, oxacillin, tetracycline, ciprofloxacin, and clindamycin in 20.9% of the isolates, followed by those isolates that were resistant to the same antimicrobials as above plus co-trimoxazole (13.4%).

### 2.2. Genotype and Phenotype Concordance Varied Depending on the Antibiotic Family

A total of 17 antibiotic resistance genes (ARGs) (Figure 1b, see Appendix A to check the MICs) were detected among the isolates, with 43.3% being phenotypically and genotypically MDR. The *bla* operon (*blaZ*, *blaI*, *blaR1*) co-occurred with all the ARGs retrieved in all isolates except for *tetM*. Additionally, the presence of the *mecA* gene was associated with the presence of *aac(6′)-aph(2″)*, *ant(6)-Ia*, *aph(3′)-IIIa*, *spw, tetM*, *tetK*, *lsaE*, *lnuB*, *ermB* and *sat-4* (Figure 2).

In general, a good correlation was observed between phenotypic and genotypic results. The concordance between the genotype and the phenotype was higher than 90% for lincosamides, trimethroprim, tetracyclines, macrolides, fluoroquinolones, and ampicillin (Appendix A). Penicillin and oxacillin showed a concordance of 88.8% and 85%, respectively. For ceftriaxon this value was 60%. However, aminoglycosides showed only 25% concordance. Moreover, when correlating phenotype and genotype, some inconsistencies were observed (Appendix A). One isolate (H_SP249, from Dog 6) was phenotypically resistant to tetracycline, but none of the published genes were detected. Another isolate (H_SP283, from Dog 8) exhibited resistance to two antimicrobial families, macrolides and beta-lactams, but in addition to presenting *ermB* and *mecA*, some genes conferring resistance to aminoglycosides were also present (*aad(6)* and *aph(3′)-IIIa*). Finally, six isolates (H_SP255, H_SP258, H_SP260 to H_SP262, and H_SP267) phenotypically susceptible to methicillin harbored the *mecA* gene.

### 2.3. Beta-Lactams and Cephalosporin Resistance

MIC values were evaluated for four beta-lactams, including penicillin, ampicillin, ceftriaxone, and oxacillin, and resistance was observed in 55, 40, 25 and 34 isolates, respectively (Appendix A).

Bioinformatic analyses detected the presence of two genes that can explain the resistance to these antibiotics: *blaZ* and *mecA*. All isolates resistant to penicillin (*n* = 55) harbored *blaZ* gene. However, seven extra isolates yielded the whole *bla* operon, including *blaZ*, *balrR1*, and *blaI* but were phenotypically susceptible to penicillin. The *mecA* gene was present in 40 isolates (Appendix A). All of them were resistant to ampicillin; 34 to oxacillin, and 25 to ceftriaxone. Interestingly, all the isolates showing the *mecA* gene and that were susceptible to both ceftriaxone (*n* = 15) and oxacillin (*n* = 6) presented the SCC*mec* type IVg (2B). When comparing the oxacillin and ceftriaxone MIC values between different SCC*mec* types, isolates (*n* = 17) with the type IVg (2B) exhibited MICs ≤1 mg/L and ≤8–16 mg/L, respectively, while the strains harboring the subtype Vc (5C2&5) (*n* = 8) or with no SCC*mec* cassette (*n* = 15) exhibited MIC values for oxacillin and ceftriaxone of >8 mg/L and >64 mg/L, respectively (Appendix A).

### 2.4. Fluoroquinolone Resistance

Phenotypic resistance to ciprofloxacin was observed in 23 isolates (13 from Dog 1 and 10 from Dog 6) and intermediate in two (Dog 2) (Appendix A). Mutations in *gyrA* and *parC* (also known as *grlA*) genes have been previously related to fluoroquinolone resistance [12]. Among the 67 *S. pseudintermedius* isolates, 38-point mutations in the *gyrA* gene were identified (Appendix A). Within these point mutations, 17 have been previously described [12], and 21 are described herein. Two of the point mutations described among our strains were specific for all the phenotypic antibiotic-resistant isolates: C251T (Ser84Leu) and G2611A (Glu871Lys). For *parC* gene, 36 point mutations were found among the 67 isolates, including 11 previously described [12] and 25 that are described herein (Appendix A). One mutation was specific for all the phenotypic-resistant isolates, G239T (Ser80Ile); however, we described six-point mutations present in 15 of the 23 resistant isolates, and three-point mutations present in three of the resistant strains.

### 2.5. Vancomycin Resistance

One isolate (H_SP204) exhibited resistance to vancomycin (MIC 16 μg/mL), and the MIC for teicoplanin was 16 μg/mL (Appendix A). None of the *van* genes conferring resistance to vancomycin were detected. However, mutations in *rpoB* and *clpP* genes were also assessed (Appendix A, respectively). Two different mutations were observed for the *rpoB* gene, encoding the β subunit of RNA polymerase, G742A (Val248Ile) and G3094A (Val1032Ile). Additionally, one mutation C17T (Thr6Ile) in the *clpP* encoding for an ATP-dependent protease involved in cell wall synthesis was found.

### 2.6. Macrolide and Lincosamide Resistance

A total of 28 isolates were phenotypically co-resistant to erythromycin and clindamycin. However, one isolate was resistant only to erythromycin (H_SP283) and another isolate to clindamycin (H_SP204) (Appendix A).

All the isolates resistant to erythromycin and clindamycin harbored the *ermB* gene, except for the clindamycin susceptible isolate. Additionally, five isolates also yielded the *lsaE* gene along with *lnuB*. *lsaE*, and *lnuB* were located next to each other, with one site-specific recombinase upstream and downstream (Figure 3).

### 2.7. Tetracycline Resistance

A total of 48 isolates were phenotypically resistant to tetracycline (47 with MICs ≥ 16 µg/mL, and one with MIC 4 µg/mL) (Appendix A). Genotypically only the isolates with MIC ≥ 16 µg/mL presented a gene related to tetracycline resistance, *tetM*. Moreover, eight isolates from the same dog presented *tetM* together with *tetK*; this last one located in a potential plasmid inserted in the chromosome. *tetK* was accompanied by the other two genes present in the pSP-G3C4 (MN612109.1) plasmid harboring *tetK* gene: *repC* (coverage 71.4% and identity 90.1%) and *repC* polypeptide A (coverage 99.5% and identity 99.4%).

### 2.8. Trimethoprim-Sulfamethoxazole (Cotrimoxazole) Resistance

A total of 23 isolates were phenotypically resistant to cotrimoxazole (trimethoprim/sulfamethoxazole) (Appendix A). In agreement, the acquired *dfrG* gene was present in 22 of the 23 resistant isolates (Appendix A). H_SP238 exhibited resistance to cotrimoxazole, but no related ARG was found.

Mutations in the constitutive gene *dhfr* were assessed in all isolates (resistant and susceptible), but no specific mutations were detected (Appendix A). Additionally, point mutations in the *folP* gene were further investigated. Interestingly, all the resistant strains and half of the susceptible strains (24/44, 54.6%) presented an insertion of 12 nucleotides (AGAGGAAGTCAC, from position 177 to 188), translated into four extra amino acids in the protein sequence (Arg-Gly-Ser-His) (Appendix A).

### 2.9. Aminoglycoside Resistance

For aminoglycosides, it was difficult to establish a clear correlation between genotypic and phenotypic results (Appendix A). Additionally, clinical breakpoints are only described for gentamicin. However, applying CARD and NCBI databases, five genes conferring resistance to aminoglycosides were retrieved: *aac(6′)-Ie-aph(2″)-Ia (aac(6′)-aph(2″)*), *ant(6)-Ia* (also known as *aad(6)*), *aph(3′)-IIIa*, *spw*, and *str*. Within 31 isolates, four different combinations of aminoglycoside resistance genes were identified (Appendix A).

The genes *aac(6′)-aph(2″), ant(6)-Ia, aph(3′)-IIIa*, and *spw* cooccurred in 96.6% to 100% of the isolates that also present the *bla* operon, *tetM*, *ermB*, and *sat-4* (Figure 2). The spw gene was also present in 100% of the isolates with *lsaE*, *lnuB*, and *dfrG*. *str* gene only cooccured with the *bla* operon and *tetM*.

Isolates H_SP285 and H_SP286 (both from the same dog) presented the str gene in a small contig independent from the chromosomal contig. The contig size was 4329 bp for H_SP285 and 4336 bp H_SP286; both were circular, with similar gene content (replicon, mobilization, and hypothetical proteins), and yielded a replicon rep7a_4_repD(pK214), indicating they were potential plasmids (Figure 4).

### 2.10. Genes Not Tested Phenotypically: sat-4 and cat

As shown in Appendix A, a total of 24 isolates presented the gene *sat-4* (encoding for a streptothricin acetyltransferase [13]) with coverage and identity values equal or higher than 99% However, five isolates from dog 5 only presented a fragment of this gene (289 bp of 543 bp) located in the negative strand of the chromosome. Analyzing the genetic context of these five isolates (H_SP224 to H_SP228) (Figure 3) and comparing it to five isolates that presented the complete gene, we observed that in all cases *sat-4* is flanked upstream and downstream by *aph(3′)-III* and *ant(6)-I*, respectively.

Sixteen isolates harbored a cat gene encoding for a chloramphenicol o-acetyltransferase in the chromosome (Appendix A). The gene was surrounded by several elements involved in recombination, mobilization, integration, and replication (Figure 3).

## 3. Discussion

In this study, we carried out a comprehensive correlation study between the genotypic and the phenotypic data of 67 *S. pseudintermedius* isolates obtained from four anatomical regions of nine healthy dogs [14]. Long-read Oxford Nanopore whole genome sequencing was performed with the MinION Mk1C followed by de novo assembling and polishing steps. Different authors have previously studied phenotype and genotype correlations in *S. pseudintermedius* using Illumina sequencing, [15,16] describing difficulties with regards to assembling the genetic context of the antibiotic resistance genes [15]. Here, we demonstrate that this avantgarde sequencing technology is an excellent tool not only to study the presence of resistance genes but also their genetic context, including insertion sequences (IS) and repeated regions.

The average number of resistances to different antibiotic families varied among the *S. pseudintermedius* isolates from one dog compared to the other dogs, but generally, the same profile was observed for the same sampled area. The nasal region should be considered a hot spot for the transmission of antibiotic resistance to other specimens since the colonization of the nasal region of a dog owner by the same *S. pseudintermedius* colonizing its dog’s nose has already been described [17].

In our previous study [18], all the *S. pseudintermedius* isolated from the skin of six healthy dogs (*n* = 22) were methicillin-susceptible *S. pseudintermedius* (MSSP), and only two of them were MDR. In contraposition, the present work identified 27 MSSP (40%) and 40 MRSP (60%), 29 of the latter also MDR, identified in five out of the nine healthy dogs. Although the number of dogs is quite limited for performing any statistical test, it appears that the presence of the *mecA* gene in *S. pseudintermedius* isolates from healthy dogs is more prevalent than previously thought. Furthermore, a complete *mecA* gene was confirmed in isolates susceptible to oxacillin and ceftriaxone, which can hinder the therapeutic outcome in case of infection. In this regard, isolates that harbored the SCC*mec* IVg exhibited lower MIC than those with SCC*mec*_subtype-Vc(5C2&5) or lacking the cassette, as described by Worthing et al. [19] Therefore, our results confirmed that oxacillin and ceftriaxone susceptibility appears to be associated with the type of SCC*mec* harbored by the bacterium.

The emergence of methicillin-resistant *Staphylococcus* spp has led to the use of vancomycin as one of the first line antimicrobials to combat Gram positive infections [20,21]. Fortunately, we could not detect any of the vancomycin resistant genes described in the literature. However, mutations in *rpoB* and *clpP* can cause reduced susceptibility against vancomycin [15,22,23,24,25]. We described herein two possible mutations associated with these genes that should be confirmed by more precise sequencing technology. Further analysis, including morphometric studies using electron microscopy and doubling time assays, should be performed as described by Cui et al. [26] to check the cell wall thickness and multiplication rate among this resistant isolate.

One hundred percent concordance between phenotype and genotype was observed for ciprofloxacin. While C251T (Ser84Leu) has been consistently described as a characteristic mutation in the *gyrA* gene conferring resistance to fluoroquinolones in different *Staphylococcus* species [12,27,28,29,30], here we report a novel second mutation that was only present in resistant isolates, G2611A (Glu871Lys). For *parC* gene, the G239T (Ser80Ile) mutation was only present in resistant isolates.

For erythromycin, 100% concordance was observed between phenotype and genotype, with two discrepancies for clindamycin resistance. Isolate H_SP204 was clindamycin resistant and erythromycin susceptible, and no antibiotic resistance gene was described. On the other hand, isolate H_SP283 presented phenotypic resistance to erythromycin but was susceptible to clindamycin. It is possible that this isolate presented an inducible clindamycin resistance, unlike all the other strains that presented the *ermB* gene and that were resistant to both erythromycin and clindamycin. It has been described in *S. pseudintermedius* that the *ermB* gene can be constitutively or induced expressed [31]. Thus, further experiments with a double-disk diffusion testing both antibiotics should be performed to ensure clindamycin resistance, since clindamycin is one of the predilected antibiotics to use in treatment when resistance to methicillin is present. Interestingly, lsaE and lnuB genes were flanked by one site-specific recombinase (Figure 3), which could denote a previous event of mobilization and integration in the chromosome. In fact, Yan et al. described these genes in a conjugative plasmid in *Enterococcus faecium,* potentially transmissible to *Staphylococcus aureus* [32]. These analyses also identified that both genes, lsaE and lnuB were located next to an antibiotic resistance region with ant(6)-I-*sat-4*-aph(3′)-III clusters [33], as described herein.

As Tyson et al. reported in their study [15] as a limitation, no concordance between phenotype and the presence of resistance genes was observed for aminoglycosides in this study. Moreover, Wegener et al. [16] described that the most usual discrepancies relied on the *aac(6′)-Ie-aph(2″)-Ia* gene. Additionally, the lack of breakpoints for some of the drugs belonging to this antimicrobial family also hampered any possible correlations between genotype and phenotype. Furthermore, several discrepancies between CARD and NCBI databases regarding coverage and gene nomenclature for this antimicrobial class made it very difficult to compare genotypic results between isolates. Still, a cluster of aminoglycoside resistance genes was detected in our isolates, comprising *ant(6)-I* (also known as *aadE*), *sat-4*, and *aph(3′)-III* with the presence of other genes such as *ermB* and *dfrG* and cat [34,35,36]. These clusters are known as Tn5405-like elements and can be found in *Enterococcus faecalis* plasmid pRE25 [37]. Loeffler and Lloyd [38] showed the rapid evolution of *S. pseudintermedius* to multidrug resistance through the acquisition of *mecA* on a staphylococcal cassette chromosome (SCC), a large transposon (Tn5405-like element) carrying up to five resistance genes, and genome point mutations for fluoroquinolones (*gyrA/grlA*) and sulfonamide resistance.

Genotypically, only the isolates exhibiting MIC ≥ 16 µg/mL for tetracycline harbored *tetM* or *tetK.* On the other hand, isolates with MIC 4 µg/mL (also categorized as resistant) did not present any known genes, although the presence of efflux pumps cannot be disregarded. In previous studies, *tetK* has been described in small size plasmids [18,39] in contraposition to our isolates that present this gene in the chromosome. According to the genetic context, it is plausible that a pSP-G3C4-like plasmid has been integrated into the chromosome, as previously described for tetK bearing *S. aureus* pT181 plasmid [40,41]. Similarly, we observed that the cat gene was surrounded by elements associated to plasmids, including a recombinase, a replication initiation protein, and a toxin-antitoxin system (zeta, ζ, and epsilon, ε, respectively). Therefore, it appears that this plasmid has also been integrated into the chromosome. The presence of plasmids carrying a cat gene integrated into the chromosome of methicillin-resistant *S. aureus* MRSA isolated from both dogs and humans has been previously reported [34].

Finally, the genetic background encoding resistance for trimethoprim or sulfonamides could not be determined since no correlation could be detected between mutations in the constitutive dihydrofolate reductase (*dhfr*) conferring resistance to trimethoprim [42,43] or dihydropteroate synthase (*folP*) target of sulfonamides [44].

## 4. Materials and Methods

### 4.1. Bacterial Cultures

The dogs came to the Hospital Clínic Veterinari (Universitat Autònoma de Barcelona, UAB, Spain) for routine preventive medical interventions (vaccinations, deworming). Prior to vaccination, they were subjected to a general physical examination by a veterinary surgeon, in which no clinical signs or lesions were detected. A total of nine healthy dogs, belonging to different breeds, and without previous records of antimicrobial treatment were selected during a period spanning one month. Samples were obtained by rubbing sterile swabs on four different skin anatomical areas: perinasal, perioral, inguinal, and perianal. These four different skin anatomical sites were chosen to represent different types of microbial habitat within the dog: from a region with fur and mostly dry like the groin (inguinal samples) to mucocutaneous areas like the muzzle (nasal and perioral samples), and the perianal region, close to the gastrointestinal tract [45,46]. Swabs were cultured in blood agar at 37 °C for 24 h. Up to five colonies morphologically compatible with S. *pseudintermedius* (small silver colonies) were selected and sub-cultured in 3 mL of Brain Heart Infusion (BHI) media at 37 °C for 16 h. A total of 67 *S. pseudintermedius* isolates were included in this study, as shown in Table 1.

### 4.2. Antibiotic Susceptibility

Minimal inhibitory concentration (MIC) assays were carried out for 18 antimicrobial compounds, using Sensititre plates (Sensititre, Trek diagnostic Systems Inc., East Grinstead, UK) (Appendix A). For confirmation of the vancomycin resistant profile, a different Sensititre MIC plate containing vancomycin and teicoplanin was used. Antimicrobial breakpoints were those recommended by the Clinical and Laboratory Standards Institute (CLSI) for the *Staphylococcus* species [47,48]. Multidrug resistant (MDR) isolates were defined for those isolates showing phenotypic resistance to at least three antimicrobials from different classes [49].

### 4.3. DNA Extraction and Nanopore Sequencing Library Preparation

DNA was extracted with ZymoBIOMICS^TM^ DNA Miniprep Kit (Zymo Research, Irvine, CA, USA). DNA quality and quantity were determined using NanoDrop 2000 Spectrophotometer and Qubit^TM^ dsDNA BR Assay Kit (Fisher Scientific SL, Madrid, Spain). The sequencing libraries prepared approximately 200 to 400ng of DNA, which were subjected to transposase fragmentation using the Rapid Barcoding Sequencing kit (SQK-RBK004; Oxford Nanopore Technologies, ONT, Oxford, UK). Up to twelve barcoded samples were loaded in a MinION FLO-MIN106 v9.4.1 flow cell (ONT, Oxford, UK) and sequenced in a MinION Mk1B or Mk1C (ONT, Oxford, UK).

### 4.4. Bioinformatic Genome Sequencing Analysis

The fast5 files were basecalled, demultiplexed, and adapters trimmed with Guppy (v5.0.11) [50] (--dna_r9.4.1_450bps_sup.cfg) (--config configuration.cfg --barcode_kits SQK-RBK004 --trim_barcodes; min_score threshold default 60). Reads with a quality score lower than 10 were discarded. Run summary statistics were obtained with Nanoplot (v1.38.1) [51] (--N50 --fastq).

Isolates were confirmed as *S. pseudintermedius* by taxonomy assignment using the EPI2ME What’s in my pot (WIMP) workflow [52]. Genomes were de novo assembled using Flye (v2.8.3) [53] (--nano-raw --plasmids --trestle), except for HSP279 and HSP281 samples, which were de novo assembled using Flye (v2.9) (--nano-hq). Contigs were polished with Medaka (v1.4.3) [54] (medaka_consensus; -m r941_min_sup_g507). Genome completeness and contamination were assessed with CheckM (v1.1.3) (lineage_wf) [55]. Circlator (v1.5.5) was used to rotate the genomes fixing the start with the dnaA gene [56] (fixstart --min_id 70), when possible.

Multilocus sequence types (MLSTs) were determined on the basis of the *S. pseudintermedius* PubMLST database (https://pubmlst.org/organisms/staphylococcus-pseudintermedius, accessed on 21 October 2022) [57] and the MLST (v2.0) software (https://cge.cbs.dtu.dk/services/MLST/, accessed on 21 October 2022) [58].

Cluster maps were plotted using the seaborn [59] library in Python. The input files were the presence/absence matrix of antimicrobial resistant genes and antimicrobial resistance.

### 4.5. Antibiotic Resistance Bioinformatic Analysis

Genomes were analysed with Abricate [60] (v1.0.1) using the Comprehensive Antibiotic Resistance Database (CARD) and the National Center for Biotechnology Information (NCBI) databases (both updated on 21 December 2021) to determine the presence of antibiotic resistance genes (ARGs), applying a threshold of identity and coverage ≥95%. Antibiotic resistance associated with point mutations of some specific genes were identified by analyzing individual files, extracting the sequences, and aligning them with BioEdit’s [61] (v7.2.5) ClustalW Multiple Alignment tool.

To determine the genetic context of some ARGs, genome sequences were uploaded to the PAThosystems Resource Integration Center (PATRIC) [62] (v3.6.12) and genome annotation was performed online with Rapid Annotation using the Subsystem Technology tool kit (RASTtk) [63] tool. SnapGene Viewer (v5.2.4) (from Insightful Science; available at https://www.snapgene.com/, accessed on 21 October 2022) was used to visualize the annotated GenBank file and to customize the output. MDR isolates were defined as those isolates showing the presence of ARG from at least three different antimicrobial families.

When the *mecA* gene was present, the Staphylococcal chromosome cassette *mec* (SCC*mec*) was described using SCCmecFinder (v1.2) (from the Center for Genomic Epidemiology of the Technical University of Denmark; available at https://cge.cbs.dtu.dk/services/SCCmecFinder/, accessed on 21 October 2022).

## 5. Conclusions

To our knowledge, this is the first study assessing the correlation between phenotype and genotype regarding antimicrobial resistance of *S. pseudintermedius* from healthy dogs using Nanopore long-read whole genome sequencing technology. Even though we were limited by the antimicrobial cutoffs for our species, a good correlation was observed for some antimicrobials, such as ciprofloxacin (fluoroquinolone), macrolides, or tetracycline. However, for oxacillin (beta-lactam) or aminoglycosides the correlation was quite low. It is noteworthy that some isolates harbored the mecA gene and were susceptible to oxacillin and/or ceftriaxone, highlighting the importance of using sequencing technology for confirmation of phenotypic results. The presence of the mecA gene and the number of MDR isolates with integrated plasmids in their chromosome emphasizes the importance of monitoring *S. pseudintermedius* for its zoonotic potential. Since we enrolled nine dogs in our study, future projects will include a broader number of individuals and a broader number of antimicrobials tested. Moreover, we will not only focus on *S. pseudintermedius*, but also on other *Staphylococcus* species isolated by culture that were discarded during the isolation steps. Finally, more analyses of the origin of the discrepancies between the phenotype and the genotype will be included.

## Figures and Tables

**Figure 1 antibiotics-11-01625-f001:**
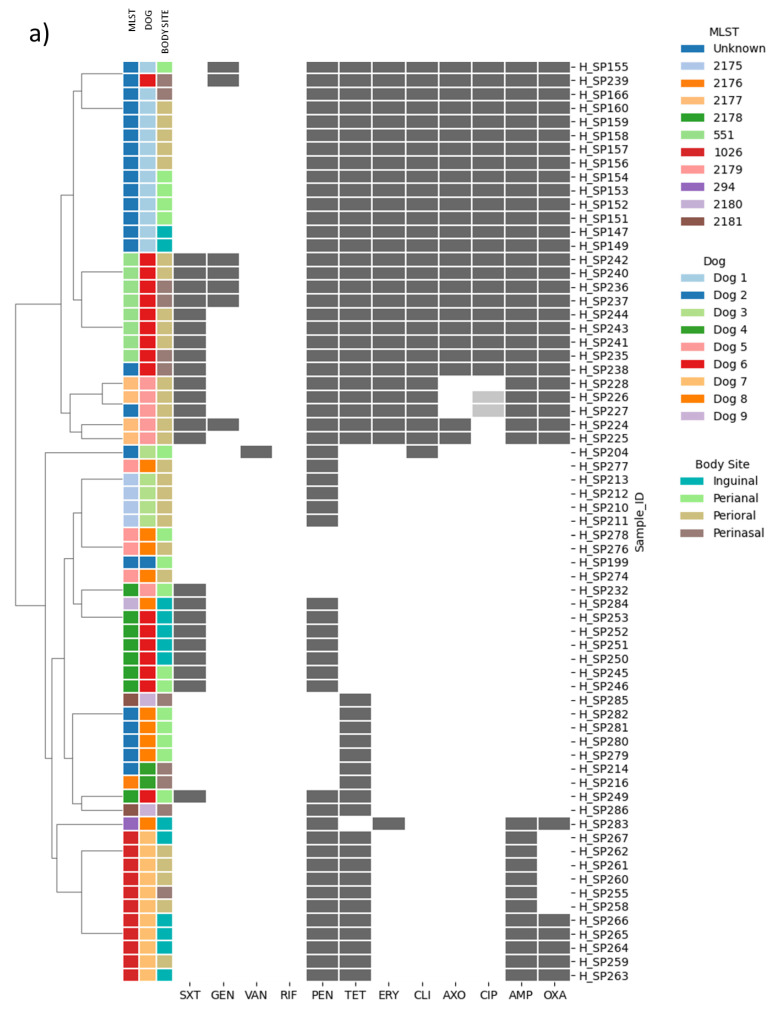
Continuation. Clustermaps for the (**a**) phenotype and (**b**) genotype results. For both clustermaps, the first three columns correspond to Multilocus Sequence type (MLST), dog number, and body site, respectively. For (**a**) phenotype clustermap, dark gray represents resistance and light gray represents intermediate susceptibility. In the (**b**) genotype clustermap, the absence/presence of antibiotic resistance determinants (genes and point mutations described in this study) are represented. SCC*mec* types are highlighted in the right side of the panel: light red for SCC*mec* subtype Vc; light green for SCC*mec* type Ivg (2B); and light blue for those strains that harbored the *mecA* gene, but no SCC*mec* was detected by https://cge.food.dtu.dk/services/SCCmecFinder/ (accessed on 21 October 2022).

**Figure 2 antibiotics-11-01625-f002:**
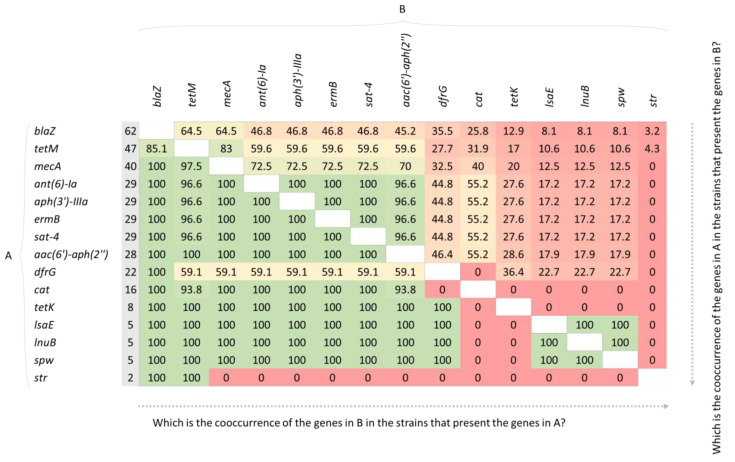
Cooccurrences (%) of the antibiotic resistance genes (ARGs). Seventeen genes were detected within the 67 *S. pseudintermedius* isolates. *blaI* and *blaR1* are not shown in the figure since they share the same presence as *blaZ*. The second column (in gray) indicates the number of isolates harboring the gene. The % values are calculated by dividing the number of cooccurrences of two given genes by the number of times a gene is present in our isolates. As an example, all the isolates presenting *mecA* (40), also present the gene *blaZ* (100% of cooccurrence); however, only 40 out of the 62 isolates presenting *blaZ* also present the *mecA* gene (64.5% of cooccurrence). For all the data and how it is calculated, see Appendix A.

**Figure 3 antibiotics-11-01625-f003:**
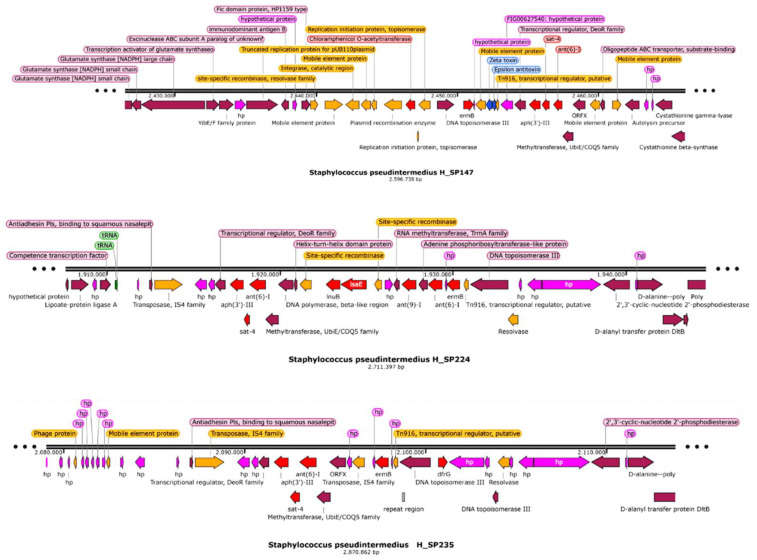
Genomic context of antibiotic resistance genes. In red, the antibiotic resistance genes; in orange, elements related to recombination, mobility, etc.; in pink, hypothetical proteins.

**Figure 4 antibiotics-11-01625-f004:**
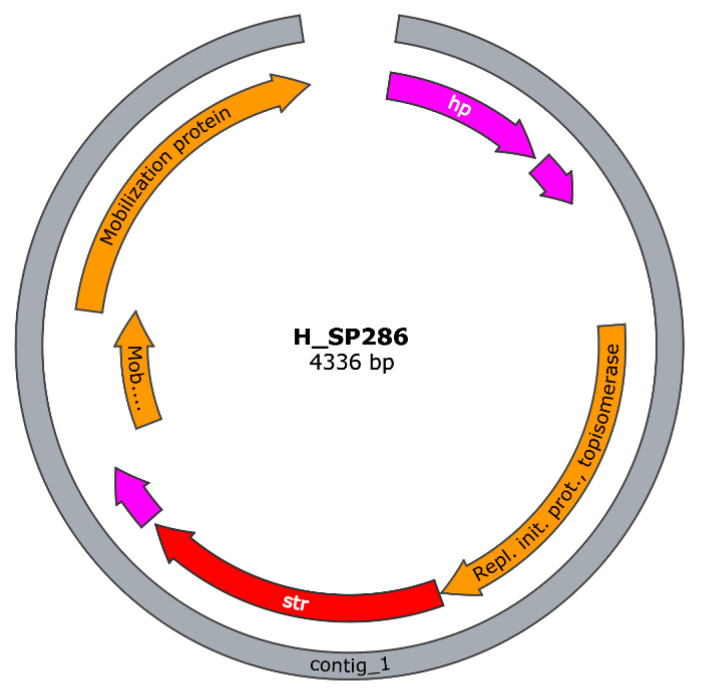
Visualization of the contig harboring str in H_SP286. The red arrow indicates the antibiotic resistance gene str; in pink, hypothetical proteins; in orange, genes related to mobilization and replication.

**Table 1 antibiotics-11-01625-t001:** Numbers and distribution of *S. pseudintermedius* isolates from four different sites in nine dogs.

Dog	Inguinal	Perianal	Perioral	Nasal	Total
1	2	5	5	1	13
2	0	1	0	0	1
3	0	1	4	0	5
4	0	0	0	2	2
5	0	1	5	0	6
6	4	3	5	5	17
7	5	0	5	1	11
8	2	5	3	0	10
9	0	0	0	2	2
Total	13	16	27	11	

## Data Availability

The genome assemblies and genomic data [14] are publicly available in GeneBank, BioProject PRJNA685966, genome accession numbers CP083183 to CP083231; CP085723, CP085724 and JAJEKF000000000 to JAJEKU000000000. The raw data are available from the Sequence Read Archive (SRA), BioProject PRJNA685966.

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
