# Peer review of "Concordance between Antimicrobial Resistance Phenotype and Genotype of Staphylococcus pseudintermedius from Healthy Dogs"

_antibiotics, 2022, doi:10.3390/antibiotics11111625_

Round 1
Reviewer 1 Report
Title: ok
Abstract:
Line 22: ..67 isolates of Staphylococcus pseudintermedius
Line 26-27: Genes are located on plasmids or chromosomes and not in them
Introduction
Line 64: Anatomical sites are more appropriate than parts
Lines 64-65: .. resistance profiles
Results
Lines 89-93: Rephrase sentence in order to bring out the meaning clearly in a coherent manner
Line 94: Table 1: The title should re-written.. Number and distribution of Staphylococcus pseudintermedius from four different sites in nine dogs.
Table 2 and Figures 1 and 3: The legend to the Table/figures should be separated from the titles and also checked for grammatical presentation.
Line 147: .. Bioinformatic analyses
Line 218: gentamicin and not gentamycin!
Discussion
Line 250: should read was performed instead of has been performed
Line 258: The average number refers to Staphylococcus isolates and not to dogs, please rephrase
Line 268: the presence of mec A was more prevalent among dog isolates of Staphylococcus pseudintermedius..
Lines 307-310: rephrase sentences
Line 315: resistance genes, not resistant genes;
Materials and methods
Line 344: the method of randomization should be stated
Line 399: de-italicize staphylococcal
Author Response
Reviewer 1:
Title: ok
Abstract:
- Line 22: ..67 isolates of Staphylococcus pseudintermedius
Thanks, we have applied the correction.
- Line 26-27: Genes are located on plasmids or chromosomes and not in them
Thanks, we have applied the correction.
Introduction:
- Line 64: Anatomical sites are more appropriate than parts
Thanks, we have applied the correction.
- Lines 64-65: .. resistance profiles
Thanks, we have applied the correction.
Results:
- Lines 89-93: Rephrase sentence in order to bring out the meaning clearly in a coherent manner
Thank you for your suggestion. We have modified the sentence; we hope that now is more understandable.
- Line 94: Table 1: The title should re-written.. Number and distribution of Staphylococcus pseudintermedius from four different sites in nine dogs.
Thanks, we have applied the correction.
- Table 2 and Figures 1 and 3: The legend to the Table/figures should be separated from the titles and also checked for grammatical presentation.
Thanks, we have applied the correction.
- Line 147: .. Bioinformatic analyses
Thanks, we have applied the correction.
- Line 218: gentamicin and not gentamycin!
Thanks, we have applied the correction.
Discussion:
- Line 250: should read was performed instead of has been performed
Thanks, we have applied the correction.
- Line 258: The average number refers to Staphylococcus isolates and not to dogs, please rephrase
Thanks, we have applied the correction.
- Line 268: the presence of mec A was more prevalent among dog isolates of Staphylococcus pseudintermedius.
Thanks, we have applied the correction.
- Lines 307-310: rephrase sentences
Thanks, we have applied the correction.
- Line 315: resistance genes, not resistant genes
Thanks, we have applied the correction.
Materials and methods:
- Line 344: the method of randomization should be stated
We have modified the statement in which we said that were randomly selected. The selection was carried out in a specific period spanning one month, and the dogs enrolled (9) were the ones that came to the clinic and that met the requirement of being healthy as described in the first paragraph of “4.1. Bacterial cultures” (new information added): “[…] they were subjected to a general physical examination by a veterinary surgeon, in which no clinical signs or lesions were detected”.
- Line 399: de-italicize staphylococcal
Thanks, we have applied the correction.
We would like to thank the editor and the reviewers for spending their time reviewing the manuscript.
Sincerely yours,
Joaquim Viñes
Reviewer 2 Report
The manuscript by Viñes et al. is complex and valuable, offering important information for the scientific community regarding the concordance between antimicrobial resistance phenotype and genotype of Staphylococcus pseudintermedius isolated from healthy dogs. I suggests some improvements, as outlined below:
Line 21: “we aimed” – I suggest to the authors to avoid the using of personal mode verb formulations, it is not so characteristic for the scientific style. Please revise these concerns throughout the manuscript!
Line 22: S. pseudintermedius strains ...
Lines 25, 407: please revise ciprofloxacin – according to my knowledge this drug belongs to the quinolones class, so, it can’t be considered class, as the authors stated!
Lines 26, 408: the same observation for oxacillin – beta lactams
Line 37: „ 92%[1].” – space delimitation, please revise this throughout the manuscript. Please improve the references at the end of the sentence, only one reference for this statement is not enough!
Line 43: „ environment[2], [3],” - incorrect citation, revise!
Line 80: I suggests rephrasing resulting in “Distribution of MRSP and MDR strains”
Line 98: “43.28” – become “43.3” -please express the results/percent with a single decimal
Line 249: “67 S. pseudintermedius isolates obtained ...”
Lines 250-251: “Long-read Oxford Nanopore whole genome sequencing has been performed with the MinION Mk1C followed by de novo assembling and polishing steps.” – seems to be from materials and methods, and not discussions section
Line 247: in the Discussion chapter the authors deeply approached the obtained results comparing with the available data from the scientific literature. However, they alternatively used either the name of the tested drugs, or the antibiotic class. Their uniformization would be useful.
Line 342: „healthy dogs” – their healthy status was previously evaluated by a veterinarian? Unclear! Why did they arrive at the hospital? – please detail
Line 344: Being an epidemiological study, I strongly suggests to the authors to justify their choice in the enrolled total number of dogs (their selection criteria, why exactly 9 and not 90, for instance). Was previously an estimated prevalence value taken in consideration? In this regard, I suggest to the authors to refer to a statistical model, based on which they can validate the study results. So, the authors must convince the scientific community that they results are completely supportable by statistical tools. It is not enough to write “... were randomly selected”
Line 345: „four different skin anatomical areas”- I wonder why? can you provide a reference?
Line 350: The authors must specify the name of the company, city and country of the used reagents
Line 357: at least three antimicrobials from different classes
Line 360: (Zymo Research) -please uniformly indicate the name of company, city, and country for the used reagents throughout the manuscript. Please revise this!
Line 394: „PATRIK” „RASTtk” – please avoid the directly using of acronyms
Line 403: the authors must highlight the study limitations and future directions/perspectives in this research area
Line 475: the complete reference list is not in agreement with the journal requirement. Please revise it!
Author Response
Reviewer 2:
The manuscript by Viñes et al. is complex and valuable, offering important information for the scientific community regarding the concordance between antimicrobial resistance phenotype and genotype of Staphylococcus pseudintermedius isolated from healthy dogs. I suggests some improvements, as outlined below:
Thank you so much for your words, we really appreciate them.
- Line 21: “we aimed” – I suggest to the authors to avoid the using of personal mode verb formulations, it is not so characteristic for the scientific style. Please revise these concerns throughout the manuscript!
Thanks, we have applied the correction.
- Line 22: S. pseudintermedius strains ...
Thank you for the suggestion. Instead of strains, we have written “isolates”.
- Lines 25, 407: please revise ciprofloxacin – according to my knowledge this drug belongs to the quinolones class, so, it can’t be considered class, as the authors stated!
Thank you for the observation. We have specified that ciprofloxacin belongs to the fluoroquinolones antibiotic class.
- Lines 26, 408: the same observation for oxacillin – beta lactams
Thank you for the observation. We have specified that oxacillin belongs to the beta-lactam antibiotic class.
- Line 37: „ 92%[1].” – space delimitation, please revise this throughout the manuscript. Please improve the references at the end of the sentence, only one reference for this statement is not enough!
Thanks. We have checked the manuscript and added a space before the reference. Moreover, we have added two more references for this statement.
- Line 43: „ environment[2], [3],” - incorrect citation, revise!
Thank you for this revision. Here we were talking about how S. pseudintermedius can get to humans through contamination or direct contact with dogs. Both references relate to that issue.
- Line 80: I suggests rephrasing resulting in “Distribution of MRSP and MDR strains”
Thank you, we have rephrased that as you suggested.
- Line 98: “43.28” – become “43.3” -please express the results/percent with a single decimal
Thanks, we have changed the percent in the text to be with a single decimal.
- Line 249: “67 pseudintermedius isolates obtained ...”
Thanks, we have applied the correction.
- Lines 250-251: “Long-read Oxford Nanopore whole genome sequencing has been performed with the MinION Mk1C followed by de novo assembling and polishing steps.” – seems to be from materials and methods, and not discussions section
Thank you for the observation. It is only a little review of the methods to introduce the discussion and reinforce the fact that we used nanopore-only sequencing to assemble and polish the genomes produced within. Further explanation about sequencing and bioinformatic analyses is described in Materials and Methods.
- Line 247: in the Discussion chapter the authors deeply approached the obtained results comparing with the available data from the scientific literature. However, they alternatively used either the name of the tested drugs, or the antibiotic class. Their uniformization would be useful.
As the reviewer has pointed out, in some points we name an antibiotic class and, in some points, we name a specific antibiotic. It depends of the context, for example in the third paragraph of the discussion we specifically talk about oxacillin and ceftriaxone because we described a specific association of these antibiotics with specific SCCmec cassette classes, so we did no generalize talking about beta-lactams. On the other hand, in the seventh paragraph we generalized and we talked about “aminoglycosides” because it was a general event among all the aminoglycoside antimicrobials tested and antibiotic resistance genes related to aminoglycoside resistance that we could not find any correlation among them.
- Line 342: „healthy dogs” – their healthy status was previously evaluated by a veterinarian? Unclear! Why did they arrive at the hospital? – please detail
We have added new information in the first paragraph of the Material and Methods section, thank you for the suggestion.
- Line 344: Being an epidemiological study, I strongly suggests to the authors to justify their choice in the enrolled total number of dogs (their selection criteria, why exactly 9 and not 90, for instance). Was previously an estimated prevalence value taken in consideration? In this regard, I suggest to the authors to refer to a statistical model, based on which they can validate the study results. So, the authors must convince the scientific community that they results are completely supportable by statistical tools. It is not enough to write “... were randomly selected”
We are sorry for the confusion, but this is not an epidemiological study, this is a descriptive study in which we assessed the concordance of the phenotypic and genotypic antibiotic resistance of Staphylococcus pseudintermedius. We have modified the statement in which we said that were randomly selected. The selection was carried out in a specific period spanning one month, and the dogs enrolled (9) were the ones that came to the clinic and that met the requirement of being healthy (new information added). All the new information is in the first paragraph of the Material and Methods section.
- Line 345: „four different skin anatomical areas”- I wonder why? can you provide a reference?
We have added new information with its respective references, thank you for the suggestion.
- Line 350: The authors must specify the name of the company, city and country of the used reagents
Done, thanks for the comment.
- Line 357: at least three antimicrobials from different classes
Applied, thanks.
- Line 360: (Zymo Research) -please uniformly indicate the name of company, city, and country for the used reagents throughout the manuscript. Please revise this!
We have stated the name of the company, city, and country for the used reagents, thank you so much.
- Line 394: „PATRIK” „RASTtk” – please avoid the directly using of acronyms
We have added the full name of the software before the acronym. We have detected in the paragraph before that CARD and NCBI were also acronyms without their meaning, so we have also added the full name. Thank you for the suggestion.
- Line 403: the authors must highlight the study limitations and future directions/perspectives in this research area
We have modified the Conclusions section adding limitations and future directions, thank you.
- Line 475: the complete reference list is not in agreement with the journal requirement. Please revise it!
Sorry about that. We have changed the reference style to the Antibiotic’s required. We have used Mendeley.
We would like to thank the editor and the reviewers for spending their time reviewing the manuscript.
Sincerely yours,
Joaquim Viñes
Reviewer 3 Report
The article "Concordance between antimicrobial resistance phenotype and genotype of Staphylococcus pseudintermedius from healthy dogs" explains systematically the association between phenotypic and genotypic information derived from Staphylococcus pseudintermedius from healthy dogs.
The article is well written and the reported work clearly explains the co-relation between genotypic and phenotypic factors within Staphylococcus pseudintermedius from healthy dogs.
The article can be improved further if the author considers it important in the introduction as well as results and discussion section to make it more explainatory and understandable.
Author Response
Reviewer 3:
“The article "Concordance between antimicrobial resistance phenotype and genotype of Staphylococcus pseudintermedius from healthy dogs" explains systematically the association between phenotypic and genotypic information derived from Staphylococcus pseudintermedius from healthy dogs.
The article is well written and the reported work clearly explains the co-relation between genotypic and phenotypic factors within Staphylococcus pseudintermedius from healthy dogs.
The article can be improved further if the author considers it important in the introduction as well as results and discussion section to make it more explanatory and understandable.”
Thank you so much for your review. We have applied the suggestions made by the other reviewers and we think that the manuscript has improved.
We would like to thank the editor and the reviewers for spending their time reviewing the manuscript.
Sincerely yours,
Joaquim Viñes